# GROUP RELATIVE ATTENTION GUIDANCE FOR IMAGE EDITING

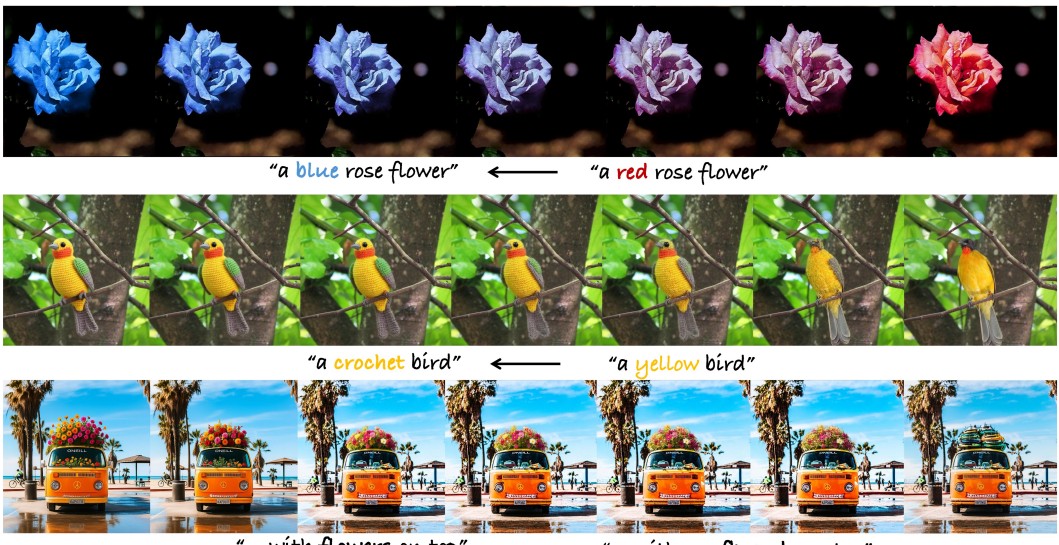

Figure 1: Variation of editing strength with respect to the relative attention guidance scale. Our method enhances attention along the intended editing direction, balancing responsiveness to instructions and consistency with the original image.

## ABSTRACT

Diffusion Transformers (DiT) have become the backbone of modern instruction-driven image editing. Yet, their capabilities remain limited by a trade-off between reference consistency and responsiveness to editing instructions. We observe a significant bias value at fixed embedding indices, which participates as a weight in the computation of attention scores. Leveraging this phenomenon, we propose Group Relative Attention Guidance (GRAG), which treats the mean of group feature vectors as a bias. By modulating the relative deviations of tokens from this bias, GRAG enhances tokens aligned with the bias direction, thereby producing more accurate editing results. Validation experiments on the latest baselines demonstrate that GRAG effectively improves the editing performance of existing models.

## 1 INTRODUCTION

Recently, Diffusion Transformer (DIT Peebles & Xie (2023)) models have once again advanced the field of text-to-image generation Black Forest Labs (2024); Esser et al. (2024). DIT employs a multimodal attention mechanism (MM-Attention) Esser et al. (2024) as its core to progressively inject semantic information from text into noisy latents, ultimately generating high-quality visual outputs through iterative denoising. Unlike UNet-based models Rombach et al. (2022); Podell et al. (2023) that separate cross-attention and self-attention, the unified attention mechanism of DiTs provides a more holistic contextual understanding. This inherent advantage enables it to perform complex image editing even without task-specific fine-tuning Avrahami et al. (2024); Wang et al. (2024). More recently, models such as Kontext Black Forest Labs (2024); Labs et al. (2025) and Qwen-

Edit Wu et al. (2025) further enhance instruction-based editing capabilities by continuing training on specialized instruction-editing datasets, demonstrating powerful controllability and generalization.

However, a persistent challenge for these instruction-based models is balancing the trade-off between maintaining fidelity to the source image and responsiveness to the editing instruction. As a result, this forces users to rely on external prompt-engineering tools or perform multiple inferences to achieve satisfactory outputs. To address this challenge, we conduct an in-depth investigation into the model's internal feature propagation, specifically how textual and visual features are integrated during the editing process. Our analysis reveals that in the MM-Attention, the token distributions of the query and key embeddings tend to cluster around a dominant bias vector, as shown in Figure 2 (a). Based on this finding, we demonstrate that by modulating the deviation of each token from this bias, it is able to achieve continuous control over the editing strength, ultimately producing controllable editing outputs.

Our investigation begins by analyzing the embedding features within each attention layer Jin et al. (2025). We identify a consistent phenomenon: within each layer, feature values concentrate around a shared bias vector. Based on the formulation of MM-Attention, this bias phenomenon can be interpreted as an intrinsic inductive pattern introduced by the architecture itself. We hypothesize that the variation of individual tokens from this bias encodes crucial contextual understanding (the theoretical analysis is presented in Section 4). This insight directly motivates our method, **Group Relative Attention Guidance (GRAG)**, a guidance mechanism also inspired by the Group Relative Policy Optimization (GRPO) Shao et al. (2024) strategy. As illustrated in Figure 2 (b), GRAG first computes the average Key embedding within each token group to determine a collective editing direction (the common bias vector). Then, a weighting coefficient $\lambda$ is used to modulate each token's $\Delta$ vector relative to the bias, enhancing those aligned with the editing intent while suppressing conflicting ones. This process leads to more precise and controllable editing outputs. We validate our method on state-of-the-art DIT-based editing models Labs et al. (2025); Wu et al. (2025); Liu et al. (2025). With a fixed guidance scale, our approach achieves a better trade-off between the editing responsiveness and image consistency, while continuous coefficient adjustment on fixed samples yields smooth and progressive editing outputs (as shown in Figure 1).

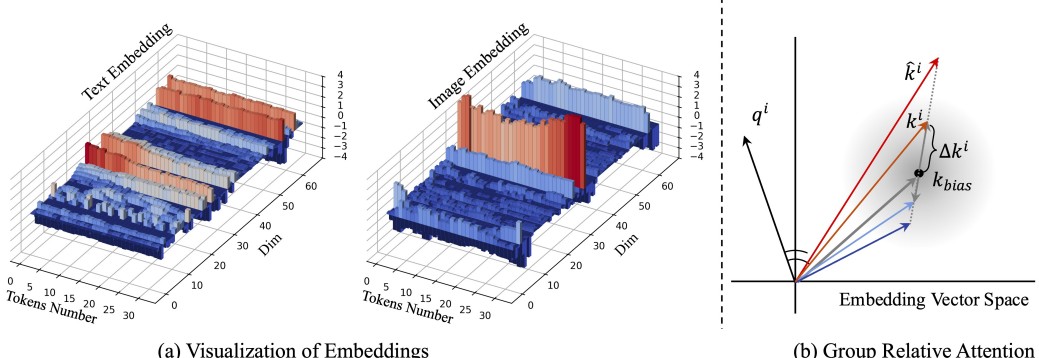

(a) Visualization of Embeddings  (b) Group Relative Attention

Figure 2: (a) presents the visualization of the embedding features input to the attention layer, where a significant bias can be observed across different tokens. (b) illustrates the proposed Group Relative Attention Guidance (GRAG), in which we amplify the differences among vectors within each group, thereby enhancing the contribution of individual vectors during editing.

Finally, our contributions can be summarized in three aspects: (a) Through extensive experiments, we identify the presence of a bias distribution in the Query and Key embeddings of MM-DIT, and we provide a mathematical analysis of its role in image editing tasks. (b) We introduce Group Relative Attention Guidance (GRAG), a novel approach that leverages the relative relationships among tokens to modulate the image editing process, enabling precise and controllable editing by modulating their deviations from the group bias. (c) We conduct extensive experiments on multiple baselines, and evaluate performance across diverse editing tasks, demonstrating the effectiveness of our method.

## 2 Related Work

### 2.1 Diffusion Transformers

Aligning textual and visual representations remains a central challenge for transformer-based diffusion models. Early work such as DiT Peebles & Xie (2023); Chen et al. (2023) replaced U-Net backbones Rombach et al. (2022) with transformers and introduced adaptive layer normalization to enable class-conditional generation, but this design limits the ability to achieve denser alignment between textual and visual information. More recent advances, such as MM-DiT Esser et al. (2024), address this limitation by introducing a unified token space and bidirectional cross-modal attention, allowing text and image tokens to interact within a shared sequence. Combined with multiple text encoders like CLIP Radford et al. (2021) and T5 Raffel et al. (2020), this design significantly enhances text understanding and enables more accurate and coherent text-guided generation. Recent studies have begun to leverage the contextual modeling capability of DiT for image editing tasks. Kontext Labs et al. (2025) adopts the same model architecture as FLUX Black Forest Labs (2024) and is further trained on instruction-based editing datasets. In contrast, Qwen-EditWu et al. (2025) replaces the T5 encoder with a large vision language model Bai et al. (2025); Liu et al. (2025) to encode both instructions and reference image information.

### 2.2 Text-Driven Image Editing with Diffusion Models.

Early works such as InstructPix2Pix Brooks et al. (2023) demonstrated that synthetic instruction–response pairs can effectively fine-tune diffusion models for image editing, while training-free methods like Textual Inversion and DreamBooth Gal et al. (2022); Ruiz et al. (2023) enabled editing with off-the-shelf generative models Rombach et al. (2022); Gal et al. (2022). Building on this foundation, subsequent editors—including Emu Edit Sheynin et al. (2024), OmniGen Xiao et al. (2025), HiDream-E1 Cai et al. (2025), and ICEdit Zhang et al. (2025)—enhanced instruction-driven editing through refined datasets and architectures, while LoRA-based methods Hu et al. (2022) introduced task-specific parameter tuning for diffusion transformers. Proprietary multimodal systems such as GPT-4V OpenAI (2023) and Gemini Team et al. (2023), along with platforms like Midjourney Midjourney (2022) and RunwayML Runway (2023), have further integrated these advances into end-to-end creative workflows. KontextLabs et al. (2025) extends the FLUXBlack Forest Labs (2024) MM-DiT model for editing tasks, leveraging its strong contextual modeling capability to achieve high consistency with reference images. In contrast, models such as Qwen-EditWu et al. (2025) enhance instruction comprehension through vision language models, enabling more complex and flexible editing operations. Despite progress, instruction-driven image editing still faces two major challenges: (i) striking a balance between editing effectiveness and consistency with the original image, and (ii) achieving precise and continuous control over editing effects. To address these challenges, we investigate the attention-layer representations of DiT and propose Group Relative Attention Guidance (GRAG), which enables precise and controllable editing effects.

## 3 Preliminaries

**Multi-Modal Diffusion Transformers.** The multi-modal diffusion transformer framework, known as multi-modal diffusion transformers (MM-DiT) Peebles & Xie (2023); Esser et al. (2024), merges both textual and visual modalities to generate images that align with the semantics of the textual inputs. FLUX incorporates a unified text-image self-attention mechanism, which aligns the multi-modal information within each MM-DiT layer. Moreover, FLUX enhances the CLIP Radford et al. (2021) text encoder by integrating the T5 Raffel et al. (2020) encoder, significantly improving its text understanding capabilities.

The MM-DiT layer uses a combined attention mechanism to fuse textual and visual data. Initially, the text tokens $T$ and image tokens $I$ are mapped into a shared space:

$$Q_\text{t} = TW_Q^\text{t}, \quad K_\text{t} = TW_K^\text{t}, \quad V_\text{t} = TW_V^\text{t}, Q_\text{i} = IW_Q^\text{i}, \quad K_\text{i} = IW_K^\text{i}, \quad V_\text{i} = VW_V^\text{i}, \quad (1)$$

where $W_Q^\text{t}, W_K^\text{t}, W_V^\text{t} \in \mathbb{R}^{d_\text{t} \times d}$ and $W_Q^\text{i}, W_K^\text{i}, W_V^\text{i} \in \mathbb{R}^{d_\text{i} \times d}$ represent the projection matrices, and $d$ denotes the shared dimension. Subsequently, the joint attention $A_\text{joint}$ is calculated by combining

the queries and keys from both the text and image modalities:

$$A_{\text{joint}} = \text{Softmax}\left(\frac{[Q_\text{t} \oplus Q_\text{i}][K_\text{t} \oplus K_\text{i}]^\top}{\sqrt{d}}\right)[V_\text{t} \oplus V_\text{i}] \tag{2}$$

where $\oplus$ denotes the token-wise concatenation of the text and image tokens. During the image editing process, the visual information consists of both the editing target and the original image: $Q_\text{i} = [Q_\text{e} \oplus Q_\text{s}]$, $K_\text{i} = [K_\text{e} \oplus K_\text{s}]$ and $V_\text{i} = [V_\text{e} \oplus V_\text{s}]$. The computation process of the corresponding attention map during editing image token update is as follows:

$$S_{edit}^{(i,j)} = \text{Softmax}(Q_\text{e}[K_\text{t} \oplus K_\text{i}])_{edit}^{(i,j)} = \frac{e^{\langle q_\text{e}^i, k_\text{t}^j \rangle}}{\underbrace{\sum_{p=1}^{N_\text{txt}} e^{\langle q_\text{e}^i, k_\text{t}^p \rangle}}_{\text{Text}} + \underbrace{\sum_{p=1}^{N_\text{img}} e^{\langle q_\text{e}^i, k_\text{e}^p \rangle}}_{\text{Editing}} + \underbrace{\sum_{p=1}^{N_\text{img}} e^{\langle q_\text{e}^i, k_\text{s}^p \rangle}}_{\text{Source}}}, \tag{3}$$

*Note:* For simplicity, the $\sqrt{d}$ is omitted.

## 4 BIAS VECTOR IN THE EMBEDDING VECTORS

The attention layer of MM-DiT serves as the key location where editing instructions and conditional image information are fused, with the query and key embeddings directly influencing the proportion of content sampled from each token. Our experiments reveal a significant bias in the distribution of embedding features along the sequence dimension, concentrated at fixed positions within each token. We hypothesize that this bias serves as a key factor in contextual understanding during the image editing process of DiT.

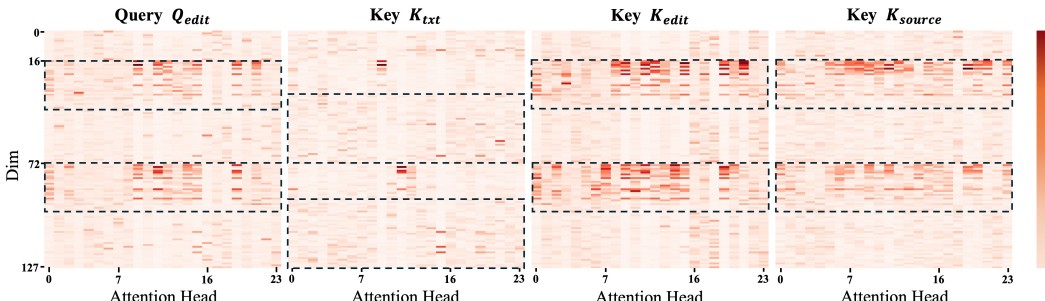

Figure 3: (Kontext-Layer 2) Aggregating different tokens along the sequence dimension, we visualize the embedding features across the dimension and head axes. The visual features are concentrated at positions corresponding to high RoPE frequencies, while textual features are associated with low frequencies.

**Concentrated distribution of embedding vectors.** For each attention layer of the transformer, we extract the query and key embeddings with shape $Q, K \in \mathbb{R}^{B \times S \times H \times D}$. For analysis purposes, we fix the batch size to $B = 1$ and partition the sequence dimension $S$ into six semantically meaningful components: $Q_\text{text}, Q_\text{edit}, Q_\text{source}$, and similarly $K_\text{text}, K_\text{edit}, K_\text{source}$. Here, $Q_\text{text}, K_\text{text} \in \mathbb{R}^{N_\text{text} \times H \times D}$ and the remaining components belong to $\mathbb{R}^{N_\text{img} \times H \times D}$. We apply $L2$ normalization along the $N_\text{text}$ or $N_\text{img}$ dimension, reducing each component to a representation in $E \in \mathbb{R}^{H \times D}$, where each element $E_{h,d}$ represents the norm of the corresponding component in head $h$ and dimension $d$. Taking $Q_\text{edit}$ as an example, $E_{h,d}$ is computed as:

$$E_{h,d} = \|Q_{:,h,d}\|_2 = \sqrt{\sum_{s=1}^{N_\text{img}} Q_{s,h,d}^2} \tag{4}$$

The visualization results of $E$ are shown in Figure 3. In the embedding vector space, each dimension index corresponds to a component, where the dark red regions in Figure 3 indicate positions with

larger magnitudes that contribute more to the inner product between different token embeddings. By examining the relationship between RoPE (Rotary Position Embedding Su et al. (2024)) and dimension indices, we observe that text embeddings concentrate in low-frequency components associated with semantics, while image embeddings concentrate in high-frequency components capturing spatial relations. This finding suggests that the two modalities are not fully aligned in the shared embedding space. Furthermore, we investigate the distribution of token embeddings in the vector space. Figure 4 presents the mean vector magnitudes and standard deviations across different attention heads, further revealing the presence of a significant bias vector among tokens in the embedding space.

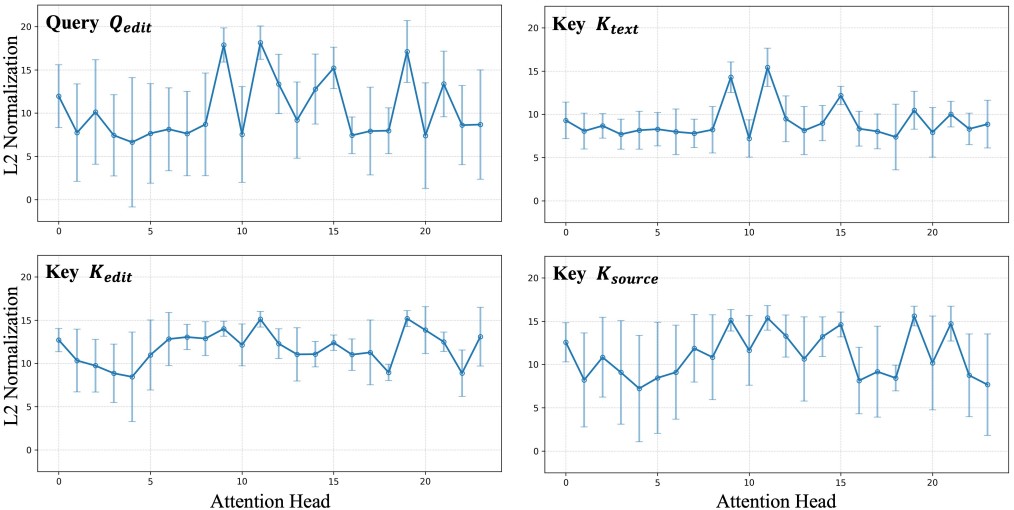

Figure 4: (Kontext-Layer 2) Mean vector magnitudes and standard deviations across different attention heads. A significant bias vector exists in the embedding space.

**Analysis of the bias vector.** The above findings suggest that the query and key embeddings in the attention layer exhibit a decomposable structure, where each can be represented as the sum of a dominant bias component and an independent variation:

$$q_i = q_{\text{bias}} + \Delta q_i, \quad k_i = k_{\text{b}} + \Delta k_i \tag{5}$$

We also observe that the feature distributions of the same layer remain highly similar across different time steps and input samples. Based on this phenomenon, we hypothesize that the bias vector $q_{\text{bias}}, k_{\text{bias}}$ is related to the model weights and represents a fixed "*editing action*" during the image editing process, while the variations of individual tokens relative to this bias vector correspond to the "*content*" being edited. Based on Equation 3, we can derive:

$$S_{\text{edit}}^{(i,j)} = \frac{e^{\langle q_e^i, (k_t^{bias} + \Delta k_t^j) \rangle}}{\sum_{p=1}^{N_{\text{txt}}} e^{\langle q_e^i, (k_t^{bias} + \Delta k_t^p) \rangle} + \sum_{p=1}^{N_{\text{img}}} e^{\langle q_e^i, (k_e^{bias} + \Delta k_e^p) \rangle} + \sum_{p=1}^{N_{\text{img}}} e^{\langle q_e^i, (k_s^{bias} + \Delta k_s^p) \rangle}}$$

$$= \frac{e^{\langle q_e^i, k_t^{bias} \rangle} e^{\langle q_e^i, \Delta k_t^j \rangle}}{e^{\langle q_e^i, k_t^{bias} \rangle} \sum_{p=1}^{N_{\text{txt}}} e^{\langle q_e^i, \Delta k_t^p \rangle} + e^{\langle q_e^i, k_e^{bias} \rangle} \sum_{p=1}^{N_{\text{img}}} e^{\langle q_e^i, \Delta k_e^p \rangle} + e^{\langle q_e^i, k_s^{bias} \rangle} \sum_{p=1}^{N_{\text{img}}} e^{\langle q_e^i, \Delta k_s^p \rangle}}, \tag{6}$$

*Note:* For simplicity, the $q_e^i$ is not extended further.

A strong shared bias component in both query and key embedding can dilute the influence of $\Delta k$, thereby reducing the sensitivity of attention scores to specific semantic differences. This insight naturally suggests that by modulating the magnitude of $\Delta k$, one can effectively control the extent to which the conditioning signals (e.g., edit instructions) influence the final output.

## 5 GROUP RELATIVE ATTENTION GUIDANCE

The variations between individual token embeddings and the bias vector reflect how the editing content relates to the current layer's *editing action*. By modulating their relative relationship, it

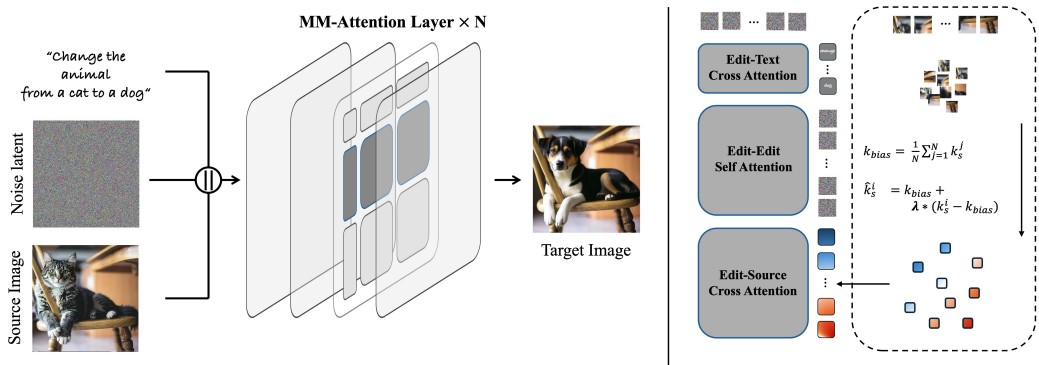

(a) Multi-Modal Image Editing Diffusion in Transformer    (b) Group Relative Attention Guidance

Figure 5: An illustration of applying Group Relative Attention Guidance in the MM-DiT image editing model. (a) The MM-Attention map corresponding to the query $Q_e$, where GRAG is applied. (b) The processing of relative modulation to the source image's key embeddings. Red denotes enhanced tokens, while blue denotes suppressed tokens.

becomes possible to achieve accurate and continuous control over the editing instructions. Based on this insight, we propose Group Relative Attention Guidance (GRAG). As illustrated in Figure 5, we modify the cross-attention component of the MM-Attention corresponding to the query $Q_e$. In Figure 5, $K_s$ is selected as a group of tokens, to which group-relative modulation is applied.

Formally, let $k_s^i$ denote the conditional key embedding corresponding to token $i$, where $i = 1, \ldots, N_{\text{img}}$. We first compute a group-level bias component as the mean of all conditional keys:

$$K_{\text{bias}} = \frac{1}{N_{\text{img}}} \sum_{j=1}^{N_{\text{img}}} k_s^j \tag{7}$$

The deviation of each token from this bias is then defined as:

$$\Delta k^i = k_s^i - k_{\text{bias}} \tag{8}$$

To control the influence of token-level variations, we introduce a tunable parameter $\lambda$ that scales these deviations:

$$\hat{k}_s^i = k_{\text{bias}} + \lambda \cdot \left( k_s^i - k_{\text{bias}} \right) \tag{9}$$

where $\hat{k}_s^i$ denotes the updated key embedding under group relative attention guidance.

The scaling factor $\lambda$ directly controls the balance between the shared bias and token-specific variations. When $\lambda > 1$, token deviations ($\Delta k_i$) are amplified, making the attention more sensitive to fine-grained differences; tokens aligned with the editing instruction are strengthened, while irrelevant ones are suppressed. Conversely, when $0 < \lambda < 1$, these variations are dampened, emphasizing the shared bias component and yielding more stable but less specific attention responses. By setting a fixed $\lambda$, the overall editing strength of the model can be controlled. Moreover, applying different scales to a given editing sample enables continuous variations in the editing results.

## 6 EXPERIMENT

### 6.1 EXPERIMENT SETTING

**Implementation Details.** We validate our proposed method against two representative baselines: Kontext Labs et al. (2025), Step1X-Edit Liu et al. (2025) and Qwen-Edit Wu et al. (2025). For reproducibility, the random seed is fixed to 42. All experiments are conducted with a batch size of 1 and 24 inference steps. The classifier-free guidance Ho & Salimans (2022) parameter is set following the recommended values for each model, 2.5 for Kontext, 6.0 for Step1X-Edit and 4.0 for Qwen-Edit. We evaluate our method on three benchmarks: PIE Ju et al. (2024), EmuEdit-Validation Sheynin et al. (2024). These benchmarks cover a diverse range of editing tasks, including

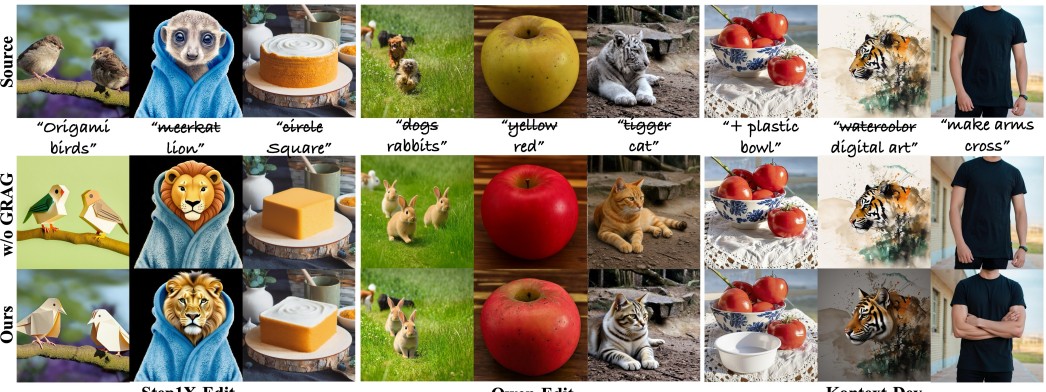

Figure 6: Visualization results on Step1X-Edit, Qwen-Edit, and Kontext.

object addition/removal, style transfer, and pose modification. For quantitative evaluation, we adopt two complementary perspectives. To assess the consistency between the edited and source images, we compute the Learned Perceptual Image Patch Similarity (LPIPS) and DINO-based structural similarity Caron et al. (2021). To measure the semantic alignment between the edited result and the editing instruction, we report CLIP similarity scores Radford et al. (2021).

## 6.2 QUALITATIVE ANALYSIS

We apply GRAG to three mainstream MM-DiT-based image editing models, with qualitative results shown in Figure 6. On Step1X-Edit and Qwen-Edit, our method improves consistency between the edited images and the original references while preserving the intended editing effects, yielding more realistic and natural outcomes. Since Step1X-Edit and Qwen-Edit leverage vision–language models to encode editing instructions, the additional instruction information often enhances responsiveness but reduces consistency. By reinforcing tokens relevant to the editing instruction and suppressing irrelevant ones, GRAG effectively mitigates this trade-off and delivers superior editing quality. For instance, in the first column of Figure 6, GRAG successfully changes the texture of the bird while retaining the details of the tree trunk; in the fifth column, it alters the color of the apple while preserving fine-grained surface details. These examples demonstrate the ability of GRAG to achieve precise and continuous control over edits while maintaining fidelity to the source image. For Kontext, our method further strengthens the influence of editing instructions on the original image, resulting in more effective and visually coherent edits.

## 6.3 QUANTITATIVE ANALYSIS

As shown in Tab 1, we evaluate the performance of GRAG relative to the baselines on multiple editing tasks from PIE and Emu. Consistent with the qualitative results, Step1X and Qwen-Edit maintain strong editing effectiveness—as reflected by nearly unchanged CLIP scores—while significantly improving consistency with the original image. Across multiple benchmarks, GRAG yields clear improvements in LPIPS, SSIM, and DINO metrics.

On Kontext, we observe a slight improvement in CLIP alignment but a relatively noticeable decline in consistency. This is because GRAG applies a fixed guidance strength to Kontext: while it enhances performance on tasks such as pose modification and style transfer, the stronger editing effect inevitably introduces more visible changes, thereby reducing similarity to the original image. We further discuss the reasons for this phenomenon in the Limitations section of the Appendix A.

## 6.4 ABLATION STUDY

**Impact of GRAG Scale on Editing Performance.** We analyze how varying the GRAG scale influences the general performance of the model. On the PIE benchmark, we test scale parameters ranging from 0.95 to 1.15. As shown in Table 2, the editing performance changes noticeably with different scale values, demonstrating the effectiveness of our method in controlling editing strength. Furthermore, the qualitative results in Figure 7 show that while the editing strength is adjusted, our method consistently preserves high-quality editing effects.

Table 1: We conduct quantitative comparative analysis on the PIE and Emu. After applying GRAG, the baseline models achieve a better balance between consistency and editing effectiveness.

| Model | PIE-Local | | | | PIE-Global | | Emu-Edit | |
|---|---|---|---|---|---|---|---|---|
| | LPIPS ↓ | SSIM ↑ | DINO ↓ | CLIP ↑ | DINO ↓ | CLIP ↑ | DINO ↓ | CLIP ↑ |
| Step1X-Edit | 0.320 | 0.900 | 0.046 | 31.32 | 0.093 | 34.39 | 0.0537 | 28.21 |
| +GRAG | **0.310** | **0.909** | **0.036** | 31.28 | **0.058** | 34.19 | **0.0373** | 28.17 |
| Kontext-Dev | 0.302 | 0.919 | 0.047 | 30.98 | 0.062 | 32.40 | 0.0533 | 28.07 |
| +GRAG | 0.342 | 0.860 | 0.062 | **31.05** | 0.082 | **32.63** | 0.0642 | 28.07 |
| Qwen-Edit | 0.339 | 0.854 | 0.059 | 31.42 | 0.086 | 33.93 | 0.0878 | 28.57 |
| +GRAG | **0.302** | **0.924** | **0.045** | 31.36 | **0.070** | **34.04** | **0.0610** | 28.57 |

**Difference With CFG.** We compare our approach with the mainstream guidance method, Classifier-Free Guidance (CFG). Unlike CFG, which adjusts the denoising direction during the denoising process, our method directly modifies the editing information within the attention layers. When the model fails to produce the desired edit under default conditions (i.e., the conditional and unconditional branches yield similar results), CFG cannot effectively adjust the editing outcome by merely amplifying the denoising direction. This limitation is reflected in Table 2, where varying the CFG strength does not lead to noticeable changes. In contrast, our method exerts precise control over the editing effects, achieving consistent and continuous adjustments as the editing strength is varied, as shown in Figura 7 Such controllability is crucial for practical editing applications.

Table 2: Quantitative analysis of different scales for CFG and GRAG.

| Method | LPIPS ↓ | SSIM ↑ | DINO ↓ | CLIP ↑ |
|---|---|---|---|---|
| CFG = 1.0 | 0.3274 | 0.8742 | 0.0486 | 31.17 |
| CFG = 2.0 | 0.32 | 0.8739 | 0.0531 | 31.31 |
| CFG = 3.0 | 0.32 | 0.8688 | 0.0567 | 31.38 |
| CFG = 4.0 | 0.33 | 0.8538 | 0.0587 | 31.42 |
| CFG = 5.0 | 0.33 | 0.8598 | 0.0589 | 31.39 |
| GRAG = 0.95 | 0.3547 | 0.8280 | 0.0641 | 31.43 |
| GRAG = 1.00 | 0.3391 | 0.8538 | 0.0587 | 31.42 |
| GRAG = 1.05 | 0.3112 | 0.9064 | 0.0453 | 31.33 |
| GRAG = 1.10 | 0.3017 | 0.9184 | 0.0444 | 31.33 |
| GRAG = 1.15 | 0.3078 | 0.9129 | 0.0448 | 31.28 |

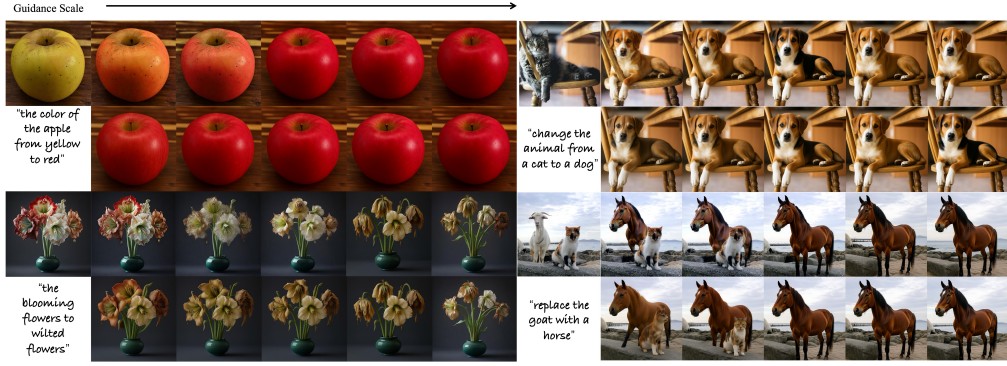

Figure 7: Visualization results of CFG and GRAG under different scales. Compared to CFG, GRAG more effectively regulates the influence of editing instructions on the original image, demonstrating a more accurate and continuous guidance process.

## 7 CONCLUSION

In this work, we investigated the limitations of DiT-based image editing models in balancing reference consistency and responsiveness to editing instructions. Through statistical analysis of the model's internal feature propagation, we identified a dominant, shared bias vector across tokens. Building on this observation, we introduce Group Relative Attention Guidance (GRAG), a method that modulates token deviations from this bias to enhance features aligned with the editing instruction. This provides finer control over the editing process, improving the balance between source image fidelity and edit accuracy. Our findings present a practical way to boost existing models and offer new insights into the internal workings of DiT-based image editing models..

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

## A  LIMITATION

While GRAG demonstrates strong effectiveness in controlling editing strength, it also has limitations. GRAG relies heavily on the accuracy of the bias vector. When applied to text tokens, the distribution of semantic tokens is less predictable, leading to uncertainty in the guidance effect. This limitation explains the weaker performance of GRAG on Kontext compared to other models.

## B  THE USE OF LARGE LANGUAGE MODELS

In this work, we use Gemini-2.5-pro to aid in the writing process. Specifically, the model was used to improve the grammatical structure, refine sentence phrasing, and enhance the overall readability of the text. The core scientific content, methodologies, and conclusions presented in this paper are the original work of the authors. The use of the LLM was restricted to a tool for language enhancement.

