# OpenReview forum: "Group Relative Attention Guidance for Image Editing"
_ICLR.cc/2026/Conference — ICLR 2026 Conference Withdrawn Submission_

### Official Review · Reviewer_vWc2 · 2025-10-30

**Soundness:** 3
**Presentation:** 3
**Contribution:** 3
**Rating:** 6
**Confidence:** 3

**Summary:**

This paper introduces Group Relative Attention Guidance (GRAG), a lightweight, tuning-free method for controlling the editing strength in Diffusion-in-Transformer (DiT) based image editing models. The core idea is based on the observation that Query and Key embeddings in the multi-modal attention layers exhibit a strong, shared bias vector. The authors hypothesize this bias represents the model's inherent editing behavior, while the deviation of each token from this bias encodes content-specific signals. GRAG works by modulating the magnitude of these deviations for a group of tokens (e.g., source image keys) before the attention calculation. This allows for continuous and fine-grained control over the balance between following the editing instruction and preserving the original image content. Experiments on several state-of-the-art DiT-based editors demonstrate that GRAG can be integrated with minimal code and provides smoother control over the editing process compared to standard Classifier-Free Guidance.

**Strengths:**

1.  **Novel Insight and Simplicity:** The paper's primary strength lies in its novel observation and interpretation of the "bias vector" within DiT attention layers. This provides a new perspective on the internal mechanics of these models. The resulting method, GRAG, is elegant in its simplicity, requiring only a few lines of code to implement, which significantly lowers the barrier to adoption.

2.  **Effective and Fine-Grained Control:** The qualitative and quantitative results (especially Fig. 1, 9 and Table 2) convincingly demonstrate that GRAG achieves a more continuous and predictable control over editing strength than varying the Classifier-Free Guidance (CFG) scale. This addresses a real and practical problem for users of instruction-based editing models.

3.  **Strong Empirical Validation:** The method is validated across multiple modern, powerful DiT-based editing models (Kontext, Qwen-Edit, Step1X-Edit), showing its applicability to different architectures. The ablation study clearly isolates the effect of the proposed guidance and its parameters (λ and δ).

**Weaknesses:**

1.  **Indirect Manipulation of Attention:** The method manipulates key embeddings *before* the attention score calculation. An arguably more direct approach to control content contribution would be to modulate the attention weights themselves (i.e., the output of the softmax operation, or the logits before it). The paper does not provide a justification for why modulating the pre-attention embeddings is a superior or more principled choice compared to these more direct alternatives.

2.  **Interpretation of the Bias Vector Lacks Direct Evidence:** The central hypothesis—that the bias vector represents "inherent editing behavior"—is intriguing but remains largely an indirect inference. The evidence provided is phenomenological (observing the bias) and functional (showing that modulating deviations from it works). The paper would be significantly strengthened by more direct evidence. For example, could the authors visualize what "action" is encoded by the bias vector alone, or correlate its properties with different categories of editing instructions?

3.  **Introduction of New Hyperparameters:** While tuning-free in the sense of not requiring retraining, GRAG introduces new hyperparameters, λ and δ. The ablation study shows that δ is the crucial one. However, it's unclear how the optimal range for δ is determined and whether it remains consistent across different models, tasks, and editing instructions. This might shift the burden from prompt engineering to "guidance scale" engineering.

**Questions:**

The interpretation of the bias vector as an "inherent editing action" is a key claim. Is it possible to provide more direct visualization or analysis of what this bias encodes? For example, what would be the output if the attention mechanism only used the mean bias vectors (i.e., setting all ∆k to zero)?

---

### Official Review · Reviewer_jQrj · 2025-10-30

**Soundness:** 2
**Presentation:** 3
**Contribution:** 4
**Rating:** 6
**Confidence:** 3

**Summary:**

This paper investigates DiT-based image editing models to balance the trade-off between fidelity to the source image and responsiveness to editing instructions. Through an analysis of the model’s internal feature propagation, it reveals that in MM-Attention, the token distributions of query and key embeddings tend to cluster around a dominant bias vector. Building on this observation, it demonstrates that modulating each token’s deviation from this bias enables continuous control over editing strength, yielding controllable editing outputs. Leveraging this insight, the paper introduces Group Relative Attention Guidance (GRAG) — a novel approach that utilizes the relative relationships among tokens to modulate the image editing process, achieving precise and controllable editing.

**Strengths:**

- Applying GRAG across various benchmarks shows that, while some trade-offs exist, the improvement in editing performance is both noticeable and acceptable. Qualitative results further demonstrate that GRAG effectively follows the given editing instructions while maintaining the source image.

- The ability to control the degree of editing by adjusting the GRAG scale highlights its high applicability. In particular, as shown in Table 2, GRAG offers more diverse and fine-grained control over editing strength and source image preservation compared to Classifier Free Guidance (CFG).

**Weaknesses:**

- While the paper identifies a significant bias vector among token embeddings and demonstrates that modulating each token’s deviation from this bias can control editing strength, the assumption stated in lines 249–252 — that the bias vector represents a fixed “editing action” during the image editing process, while the variations of individual tokens relative to this bias correspond to the “content” being edited — lacks a concrete theoretical justification and is only supported empirically.

- In lines 308–312, the authors claim that when the scaling factor is greater than 1, tokens aligned with the editing instruction are strengthened while irrelevant ones are suppressed; when the scaling factor is between 0 and 1, the shared bias component is emphasized, resulting in more stable but less specific attention responses. However, if this argument holds, increasing the scaling factor should result in stronger editing effects. Yet, in Table 2, the CLIP score decreases as the GRAG scale increases, which appears to contradict the authors’ explanation.

**Questions:**

- In lines 217–221, the authors observe that text embeddings concentrate in low-frequency components, while image embeddings concentrate in high-frequency components. When GRAG is applied only to text embeddings, does it mainly affect the low-frequency semantic information? It would be helpful to understand the distinct effects of GRAG when applied to text embeddings versus image embeddings.

- Since multiple MM-Attention layers exist in the model, it would be valuable to analyze how applying GRAG at different layers affects the image editing process. Specifically, how does applying GRAG in earlier layers versus later layers influence the resulting edited image?

---

### Official Review · Reviewer_H37y · 2025-10-31

**Soundness:** 3
**Presentation:** 2
**Contribution:** 3
**Rating:** 4
**Confidence:** 4

**Summary:**

This paper introduces GRAG, a training-free method for instruction-driven image editing in Diffusion Transformers (DiT). The key observation is that attention embeddings contain a dominant bias vector, which represents the editing action. Individual token deviations from this bias then encode the image's content. GRAG works by calculating this group-level bias and then scaling each token's deviation. This enhances tokens relevant to the edit and suppresses conflicting ones, offering fine-grained control without training. GRAG improves the balance between following instructions and preserving the source image, as shown by both visual results and metrics like LPIPS and DINO. The authors argue GRAG provides more precise control at the attention level, especially when other methods like CFG saturate.

**Strengths:**

1. The method is simple and training-free, and is supported by solid qualitative and quantitative evaluations across multiple editors, including an ablation study.
2. The method consistently produces more accurate editing results, effectively enhancing the performance of existing base models.
3. It effectively addresses the key trade-off between instruction fidelity and source preservation.

**Weaknesses:**

1. The comparison is narrow. The paper's primary comparator is CFG, but it overlooks other relevant, training-free editing controls like attention reweighting, PnP Inversion, attention injection, and token-level gating. A broader comparison is essential to substantiate the claim of "uniquely precise and continuous control".
2. The guidance coefficient λ is treated as a singular, static parameter with no exploration of task, dataset, or timestep-dependent tuning. Furthermore, no formal link is established between λ and interpretable metrics like attention entropy, which limits its explainability and principled selection.
3. The paper lacks sufficient specifics about the experimental setup.
4. The manuscript's presentation quality needs improvement. Key figures (e.g., Figure 3, Figure 5) suffer from poor legibility. Additionally, there are inconsistencies in notation and equation formatting, and the writing in Chapter 3 (Preliminaries) contains errors in punctuation and spacing that disrupt the reading flow.

**Questions:**

1. Does GRAG apply to all attention layers and heads, or only to a subset? Have you conducted ablation experiments to validate the impact of applying GRAG in specific layers/heads versus globally?
2. GRAG currently computes K_bias using the mean across the selected token group. Have you tried alternative estimators (e.g., median, trimmed mean, Huber mean, or dimension-wise robust averaging)? How sensitive are the results to outliers and the choice of estimator?
3. Is the guidance coefficient λ constant throughout the denoising timesteps, or does it vary?

---

### Note · Authors · 2025-11-14

I have read and agree with the venue's withdrawal policy on behalf of myself and my co-authors.